# Cost-effectiveness of a primary healthcare intervention to treat male lower urinary tract symptoms: the TRIUMPH cluster randomised controlled trial

Madeleine Cochrane ,[1] Marcus J Drake,[2] Jo Worthington,[1] Jessica Frost,[1] Nikki Cotterill,[3] Mandy Fader,[4] Lucy McGeagh ,[5] Hashim Hashim,[6] Athene Lane,[1] Margaret Macaulay,[7] Stephanie MacNeill,[1] Jonathan Rees,[8] Matthew J Ridd ,[1] Luke A Robles ,[9] Emily Sanderson,[1] Gordon Taylor,[10] Jodi Taylor,[1] Sian Noble[1]

**Correspondence to**
Dr Madeleine Cochrane;
madeleine.cochrane@bristol.ac.uk

## ABSTRACT

**Objectives** To estimate the cost-effectiveness of a primary care intervention for male lower urinary tract symptoms (LUTS) compared with usual care.

**Design** Economic evaluation alongside a cluster randomised controlled trial from a UK National Health Service (NHS) perspective with a 12-month time horizon.

**Setting** Thirty NHS general practice sites in England.

**Participants** 1077 men aged 18 or older identified in primary care with bothersome LUTS.

**Interventions** A standardised and manualised intervention for the treatment of bothersome LUTS was compared with usual care. The intervention group (n=524) received a standardised information booklet with guidance on conservative treatment for LUTS, urinary symptom assessment and follow-up contacts for 12 weeks. The usual care group (n=553) followed local guidelines between general practice sites.

**Measures** Resource use was obtained from electronic health records, trial staff and participants, and valued using UK reference costs. Quality-adjusted life-years (QALYs) were calculated from the EQ-5D-5L questionnaire. Adjusted mean differences in costs and QALYs and incremental net monetary benefit were estimated.

**Results** 866 of 1077 (80.4%) participants had complete data and were included in the base-case analysis. Over the 12-month follow-up period, intervention and usual care arms had similar mean adjusted costs and QALYs. Mean differences were lower in the intervention arm for adjusted costs −£29.99 (95% CI −£109.84 to £22.63) while higher in the intervention arm for adjusted QALYs 0.001 (95% CI −0.011 to 0.014). The incremental net monetary benefit statistic was £48.01 (95% CI −£225.83 to £321.85) at the National Institute for Health and Care Excellence UK threshold of £20 000 per QALY. The cost-effectiveness acceptability curve showed a 63% probability of the intervention arm being cost-effective at this threshold.

**Conclusions** Costs and QALYs were similar between the two arms at 12 months follow-up. This indicates that the

## STRENGTHS AND LIMITATIONS OF THIS STUDY

⇒ This was a multicentre study that collected data from a large number of general practice sites and participants.
⇒ A strength of using electronic health records was that it reduced the data collection and recall burden for participants.
⇒ A limitation of using electronic health records was that different levels of detail were provided by the two general practice IT systems.

intervention can be implemented in general practice at neutral cost.

**Trial registration number** ISRCTN11669964.

## INTRODUCTION

Lower urinary tract symptoms (LUTS) is a bothersome condition among the ageing male population with a prevalence of up to 30% in men over 65 years old.[1] LUTS includes storage, voiding and postmicturition symptoms and can have a considerable impact on quality of life.[2] Causes of LUTS include benign prostate enlargement, detrusor muscle weakness or overactivity, prostate inflammation and lifestyle habits. Evidence-based national guidelines from the UK's National Institute for Health and Care Excellence (NICE) recommend conservative management interventions (such as fluid and caffeine intake, bladder training, urethral compression and release, and pelvic floor muscle exercises) at initial assessments for LUTS, once serious underlying conditions (eg, urinary tract infection, prostate cancer and neurological disease) have been excluded.[1]

Conservative management options[3] are time-consuming and complex to implement due to the range of possible symptoms and causes of LUTS. Initial evidence from a small single centre randomised controlled trial (RCT) suggests conservative self-management interventions could play a role in the delivery of treatment for LUTS. However, this smaller trial only evaluated effectiveness and not cost-effectiveness.[4] The present study, TReatIng Urinary symptoms in Men in Primary Healthcare (TRIUMPH), sought to provide evidence on whether the provision of a manualised and standardised non-pharmacological intervention in primary care, is effective and cost-effective compared with usual care. The effectiveness results, reported elsewhere, showed improved symptoms for men with LUTS and these benefits were sustained for a full year.[5] The current cost-effectiveness study is conducted from a National Health Service (NHS) perspective over a 12-month time horizon, with the intention to inform clinical practice decisions within primary care. The analysis is reported according to the Consolidated Health Economic Evaluation Reporting Standards guidelines.[6]

## METHODS
### Target population and setting
Participants in the TRIUMPH study were men aged 18 or over, who considered themselves to have bothersome LUTS and had presented to primary care within the past 5 years with at least one symptom of LUTS. The trial was carried out across 30 general practice (GP) sites in Southwest England.

### Comparators
The intervention arm, a manualised and standardised intervention, involved the provision of an information booklet with guidance on conservative interventions for LUTS. Participants had an initial symptom assessment with a nurse or healthcare assistant (HCA) in order to understand their symptom needs, bothersomeness and impact on their quality of life, as well as their personal circumstances. This was delivered face to face. Participants were then directed to specific sections of the information booklet. Participants were offered up to three further contacts from the healthcare professional. The first of the three was offered via telephone while for the remaining two, participants were given the option to receive these via telephone, email or text. The comparator group continued to receive care as usual.

### Study design
The economic evaluation was conducted alongside the TRIUMPH trial, a multicentre cluster RCT and evaluated costs and outcomes from an NHS perspective over a 12-month time horizon. Detailed methods for the design of the TRIUMPH trial (ISRCTN11669964) are described elsewhere.[7] In brief, 30 GP sites were randomised on a 1:1 basis to either deliver the intervention or to continue care

as usual. Randomisation was minimised by NIHR Clinical Research Network centre (West of England and Wessex), practice size and area-level deprivation.

### Patient and public involvement
A patient advisory group (PAG) was set up to ensure patient and public involvement (PPI) throughout all stages of the research. PPI representatives were involved in the design of the study, ensuring all study material were usable and appropriate for a non-specialist audience. In particular, the PPI group reviewed and advised on the following study components: TRIUMPH intervention booklet, patient-facing questionnaires, newsletters, website, initial qualitative results relating to men's experiences of the patient pathways for LUTS within the NHS and agreeing to plans for implementing and disseminating the study results. Two PPI representatives also played a significant role in planning and chairing the PAG meetings. In addition, both the trial management group and trial steering committee each included a patient representative who contributed to the management and conduct of the trial.

### Outcomes
The quality-adjusted life-year (QALY) was used as the outcome measure, as recommended in the UK's reference case[8] for interventions funded by the NHS. Health status was measured using the EuroQol 5-Dimension 5-Level (EQ-5D-5L) questionnaire and EuroQol Visual Analogue Scale (EQ-VAS) at baseline, 6 and 12 months. At baseline, the questionnaire was administered via post, while at 6 and 12 months, it could be completed by post or online. Participants' EQ-5D-5L scores were mapped to the EQ-5D-3L valuation set using a validated mapping function by van Hout et al.[9] This enabled a utility score to be calculated for each participant based on published pre-existing utility scores derived from a representative sample of the UK population. Utility scores were then combined with length of life to calculate the QALYs for each participant using an area-under-the-curve approach.[10] The approach accounted for any deaths that occurred during the 12-month trial period.

### Resource use data collection and valuation
Resource use data were obtained for intervention training and delivery costs, primary care consultations and LUTS-related medication and secondary care. Data for each individual patient's 12-month follow-up period was captured via the following sources: trial management records, case report forms, primary care electronic health records (EHR) data and self-report questionnaires.

Intervention-related resources in terms of training, including staff time, travel and teleconference expenses, were obtained primarily through trial management records. Expert opinion from the lead nurses was used to capture duration of the training. Resources used in the delivery of the intervention in terms of staff time for the initial clinic visit and subsequent follow-up were prospectively recorded on case report forms. Primary

care EHRs were used to capture: (1) all primary care consultations with a GP, nurse, HCA or pharmacist; and (2) LUTS-related prescribed medication. Database searches were built to extract EHRs from primary care administration systems (EMIS and SystmOne). Records were extracted from each GP site at least one month after the last recruited participant had reached their 12-month follow-up date.

Secondary care was predominantly collected via patient self-report questionnaires at 6 and 12 month follow-up, administered either online or via post. Participants were asked to report LUTS-related outpatient, day case, inpatient and A&E visits. EHR data were used to identify secondary care referrals. If a participant had a urology referral but had missing or no secondary care self-report data, then we reviewed the participant's secondary care letters.

A urology clinician was consulted to identify LUTS-related prescribed medications and secondary care visits. More specifically, prescribed medications were extracted from the EHRs if they were coded under the drug criteria: urinary frequency, nocturnal enuresis, incontinence, urinary retention or diuretics. The clinician reviewed the names and doses of the extracted medications and created a list of LUTS-related medications which was applied by the analyst. Similarly, the clinician reviewed a list of the types of secondary care visits reported by the participants and in the primary care letters. The clinician created a list of LUTS-related care and this was applied by the analyst.

Details of how the resources were valued are reported in table 1. 2018/2019 costs were used to value the resource use. The NHS cost inflation index was used to inflate a unit cost to current prices if it was unavailable for the year of analysis.[11]

## Analysis

An analysis plan was prespecified before all data were collected and STATA V.16.1 was used for all analyses.[12] The primary economic analysis was to evaluate costs and outcomes over a 12-month time horizon and so discounting was not required. EHR data were derived from two different Information Technology (IT) systems, a standardised cleaning process was, therefore, implemented to ensure the final analysis dataset was comparable across the two systems. All participants were analysed in accordance with the allocation group they were assigned to at the point of randomisation.

Simple mean imputation was used for a minority of participants who had missing data in relation to their length of follow-up visit at week 1 (n=1) and week 4 (n=7) in their case report forms. Additionally, it was assumed zero secondary care visits occurred if other parts of the questionnaire had been completed but the resource use questions had been left blank and the patient had no urology referrals recorded in their EHRs. Total costs were then calculated for each individual participant by summing all the costs across all categories (intervention, primary care, LUTS medications and LUTS secondary care) and mean total unadjusted costs per arm were then estimated.

A mixed effects multilevel model (MLM) using random intercepts and assuming a two-level structure was used to account for clustered data and to adjust for the minimisation variables (practice size, Index of Multiple Deprivation (IMD) and centre) as well as GP IT system and baseline utility for costs and QALYs,[13] respectively. El Alili *et al*[14] found MLMs performed better than OLS regression for cluster-based economic evaluations. The MLM was used to calculate the adjusted mean costs and QALYs, and the adjusted mean differences in costs and QALYs between groups. For both costs and QALYs, 95% CIs were estimated using bias-corrected and accelerated bootstrapping.[14]

Regression outputs from the MLM facilitated the estimation of an incremental net monetary benefit (INMB) statistic and the associated CIs at NICE's recommended willingness to pay threshold of £20000 and £30000 per QALY (MLM outputs in tables 1 and 2, (online supplemental material). A cost-effectiveness acceptability curve (CEAC) was constructed to present the uncertainty over a range of willingness to pay thresholds. This was done using the parametric p value approach as applied by El Alili *et al*[14].

One-way sensitivity analyses were conducted to explore methodological uncertainty:

► An alternative analytical model was employed by using a seemingly unrelated regression (SUR) model instead of an MLM.
► Inconsistency in coding the mode of delivery for primary care consultations was explored by recoding all unit costs for telephone contacts to unit costs for face-to-face contacts.
► Training costs were excluded from the total intervention costs, as it is possible that these costs are captured in the published unit costs.
► Intervention training and delivery costs were excluded.
► The primary analysis model was adjusted for GP consultation costs for the 12-month period preconsent.
► Missing data assumptions were tested using a range of approaches, including:
  a. For our primary analysis, we assumed our data were missing completely at random. In addition, we made the assumption that our data were missing at random and applied multiple imputation by chained equations with predictive mean matching and our SUR model. The covariates in the multiple imputation model were site ID, baseline utility, centre, practice size, IMD and GP IT system. The model was run by allocation group and a randomisation seed was set to ensure reproducible imputations. A total of 25 imputations were performed and combined with Rubin's rule in Stata V.16.1.[15]
  b. For each arm, zero costs and a QALY value that was 10% higher than the mean QALY value were

**Table 1** Resources collected and their valuation

| Resources | Unit cost (£) | Source of unit cost |
|---|---|---|
| GP surgery visit | 34[*][†] | Curtis and Burns, 2019[11] |
| GP telephone call | 26.27[*][†][‡] | Curtis and Burns, 2019[11] |
| GP home visit | 87.28[*][†][§] | Curtis and Burns, 2019[11] |
| Practice nurse surgery visit | 10.85[†][¶] | Curtis and Burns, 2019[11] |
| Practice nurse telephone call | 8.25[†][**] | Curtis and Burns, 2019[11] |
| Healthcare assistant surgery visit | 6.67[¶] | Curtis and Burns, 2019[11] |
| Healthcare assistant telephone call | 5.07[**] | Curtis and Burns, 2019[11] |
| Pharmacist surgery visit | 11.63[¶] | Curtis and Burns, 2019[11] |
| Pharmacist surgery telephone call | 8.84[**] | Curtis and Burns, 2019[11] |
| NHS 111 call | 12.26 | Pope et al, 2017[23] |
| Trainers (professor; lead nurses) | Varies | Research Institution's pay scales, 2018/2019[24]; Curtis and Burns, 2019[11] |
| Intervention booklet | 6.15[§§] | Research Institution's printing service charge[25] |
| Intervention visit time per minute | Varies[¶¶] | Research Institution's pay scales, 2018/2019[††],[24] Curtis and Burns, 2019[‡‡][11] |
| Follow-up contact time per minute | Varies[***] | Curtis and Burns, 2019[11] |
| Teleconference call per minute | 0.15 | BT Conference Call service charge[26] |
| Car mileage | 0.45[†††] | HM Revenue and Customs[27] |
| Outpatient procedures | Varies | NHS National Cost Collection, 2018/2019[28] |
| Outpatient visits | 110[‡‡‡] | NHS National Cost Collection, 2018/19[28] |
| Day cases | Varies | NHS National Cost Collection, 2018/2019[28] |
| Inpatient admissions | Varies | NHS National Cost Collection, 2018/2019[28] |
| Accident & Emergency attendances | 159 | NHS National Cost Collection, 2018/2019[28] |
| Medications | Varies[§§§] | Precription Cost Analysis, 2019[29];NHS Drug Tariff, 2019[30] |

*Excluding direct care staff costs.
†Including qualifications.
‡Based on the assumption of a 7.1 min telephone consultation, as reported in earlier unit cost series by Curtis and Burns.[31]
§Based on the assumption of a 11.4 min home visit and 12 min of travel, as reported in earlier unit cost series by Curtis and Burns.[31]
¶Based on the assumption of a 15.5 min surgery consultation, as reported for a practice nurse in earlier unit cost series by Curtis and Burns. It was assumed the Nurses (practice and research), HCAs and pharmacists had the same surgery consultation length.[31]
**There is no published unit cost for non-triage telephone consultations for practice nurses, healthcare assistants and pharmacists. Telephone consultations were assumed to be 11.8 min for these healthcare professionals. This was based on data in Curtis and Burns 2015 and 2019 series, which indicates face-to-face and triage telephone consultations delivered by a practice nurse are approximately 1.5 times longer than those provided by a GP.[11][31]
††Including basic salary, national insurance and superannuation.
‡‡Assuming Band 6 for all lead nurses.
§§All booklets had to be prepared in advance, it was, therefore, assumed all participants randomised to the intervention arm incurred the booklet cost.
¶¶For intervention visits, length of visit was reported in minutes in the case report form. Unit cost varied depending upon which staff type delivered the intervention.
***For telephone follow-ups, length of visit was reported in minutes in the case report form. Unit cost varied depending upon which staff type delivered the intervention. For texts and emails, a number of these were reported in the case report form. Time taken to send each text and/or email was assumed to be 2 min.
†††Distance travelled was calculated by measuring the distance between the staff's workplace and the training locations.
‡‡‡For outpatient visits a weighted average of the unit costs for consultant and non-consultant led visits was used.
§§§In the minority of instances where a unit cost was not available in the prescription cost analysis, then the unit costs from the 2019 NHS Drug Tariff were used.
GP, General Practitioner; NHS, National Health Service.

**Table 2** Resource use and costs from NHS perspective

| Resource category | Intervention (n=524) | | | Usual care (n=553) | | |
|---|---|---|---|---|---|---|
| | n | Resource use, mean (SD) | Cost, mean (SD) (£) | n | Resource use, mean (SD) | Cost, mean (SD) (£) |
| GP consultations all types | 478 | 4.60 (4.67) | 145.86 (162.62) | 544 | 5.00 (5.68) | 158.17 (180.05) |
| Practice nurse consultations all types | 478 | 2.78 (3.64) | 28.77 (38.58) | 544 | 3.61 (5.75) | 38.41 (58.48) |
| HCA all types of consultations | 478 | 1.16 (2.05) | 7.70 (13.67) | 544 | 0.59 (2.07) | 3.91 (13.82) |
| Pharmacist all types of consultations | 478 | 0.15 (0.54) | 1.60 (5.59) | 544 | 0.15 (0.90) | 1.36 (7.18) |
| Total LUTS medications | 478 | 3.27 (4.74) | 45.77 (100.50) | 544 | 4.29 (5.75) | 42.22 (98.97) |
| NHS 111 encounters | 459 | 0 | 0.00 | 479 | 0.01 (0.14) | 0.08 (1.72) |
| Outpatient visits | 459 | 0.07 (0.34) | 8.04 (36.75) | 479 | 0.08 (0.37) | 8.96 (40.82) |
| Outpatient procedures | 459 | 0.03 (0.20) | 4.00 (27.78) | 479 | 0.05 (0.31) | 6.43 (39.65) |
| Inpatient stay | 459 | 0 | 0.00 | 479 | 0.00 (0.09) | 7.10 (155.49) |
| Accident and emergency visits | 459 | 0.00 (0.09) | 0.69 (14.84) | 479 | 0.01 (0.08) | 0.99 (12.56) |
| Total intervention delivery costs | 524 | | 31.48 (10.16) | – | | 0.00 |
| Total intervention training costs | 524 | | 7.66 | – | | 0.00 |
| Total adjusted primary care consultation costs* | 478 | | £174.48 (£142.80 to£206.16) | 544 | | £205.61 (£166.05 to£245.18) |
| Total adjusted medication costs* | 478 | | £40.49 (£28.28 to £52.71) | 544 | | £46.39 (£34.90 to £57.88) |
| Total adjusted secondary care consultation cost*† | 459 | | £13.78 (£0.37 to £27.19) | 479 | | £25.83(£0.06 to £51.59) |

*Adjusted for centre, practice size, area-level deprivation and general practice Information Technology (IT) system.
†Zero-day case visits were reported in either arm.
GP, General Practitioner; HCA, Healthcare Assistant; LUTS, Lower Urinary Tract Symptoms; NHS, National Health Service.

imputed for participants with missing questionnaire data.

c. For participants with missing questionnaire data, the largest participant cost for secondary care and a QALY value 10% lower than the mean QALY value per arm were imputed.

d. SystmOne EHR-based practices did not include data on the quantity of tablets prescribed per medication. In the base-case analysis, EMIS practices data were used to estimate the mean quantity in the SystmOne practices. A sensitivity analysis used the most frequently reported quantity, rather than the mean.

A post hoc subgroup analysis compared participants who had completed follow-up from 11th March 2020 (where 11 March 2020 reflects when the COVID-19 outbreak was declared internationally as a pandemic) to those who completed follow-up before this date.

## RESULTS

A total of 1077 men were recruited; either to receive the intervention (n=524) or usual care (n=553). There was a low rate (5.1%, n=55) of missing EHR data (primary care and medications), and a greater rate of item missingness for the self-report secondary care (12.5%, n=135) and EQ-5D-5L (16.1%, n=173) data. This rate of missing data permitted a complete-case primary analysis on 866 out of 1077 (80.4%) participants, 413 (78.8%) and 453 (81.9%) participants in the intervention and usual care arm, respectively.

### Resource use, costs and outcomes
Table 2 shows that mean resource use and costs from available cases were similar between arms, as were the adjusted mean costs for total primary care consultations, medications and secondary care visits. Mean intervention costs were £39, excluding intervention training costs reduced this to £31. Overall, EQ-5D-5L scores, QALYs and EQ-VAS scores were similar across both groups and all time points (table 3).

### Cost-effectiveness analysis
As shown in table 4, over the 12-month analysis period intervention and usual care arms had similar mean

**Table 3** Mean unadjusted EQ-5D-5L health index score, QALYs and EQ-VAS score by trial arm

| Time period | Intervention | | Usual care | |
|---|---|---|---|---|
| | Mean (SD)* | n | Mean (SD)* | n |
| EQ-5D-5L health index score and QALYs | | | | |
| Baseline | 0.83 (0.17) | 522 | 0.83 (0.16) | 547 |
| 6 months | 0.84 (0.17) | 483 | 0.84 (0.16) | 509 |
| 12 months | 0.83 (0.17) | 457 | 0.83 (0.17) | 482 |
| Total QALYs | 0.84 (0.16) | 444 | 0.84 (0.15) | 460 |
| EQ-VAS score | | | | |
| Baseline | 78.05 (15.27) | 522 | 77.81 (15.49) | 551 |
| 6 months | 77.87 (15.93) | 481 | 76.34 (16.30) | 505 |
| 12 months | 77.13 (16.48) | 463 | 76.11 (16.46) | 481 |

*Higher scores represent better health.
EQ-5D-5L, EuroQol 5-Dimension 5-Level questionnaire; EQ-VAS, EuroQol Visual Analogue Scale; QALYs, quality-adjusted life-years.

**Table 4** Primary analysis and sensitivity analyses

| Trial arm | n | Adjusted*, mean (95% CI)† | | Incremental adjusted* mean (95% CI)† | | |
| | | Costs (£) | QALYs | Costs (£) | QALYs | INMB (£) at £20 000/QALY (95% CI) |
| --- | --- | --- | --- | --- | --- | --- |
| Primary analysis: complete-case analysis with MLM | | | | | | |
| Intervention | 413 | £253.53 (£215.85 to £291.22) | 0.836 (0.828 to 0.845) | −£29.99 (−£109.84 to £22.63) | 0.001 (−0.011 to 0.014) | £48.01 (−£225.83 to £321.85) |
| Usual care | 453 | £283.52 (£235.40 to £331.64) | 0.836 (0.828 to 0.843) | | | |
| Alternative analysis: complete-case analysis with SUR | | | | | | |
| Intervention | 413 | £258.05 (£229.90 to £286.21) | 0.836 (0.829 to 0.844) | −£20.00 (−£62.50 to £22.50) | 0.001 (−0.010 to 0.011) | £35.54 −£182.20 to £253.28) |
| Usual care | 453 | £278.05 (£251.42 to£304.68) | 0.836 (0.829 to 0.842) | | | |
| Multiple Imputation with SUR | | | | | | |
| Intervention | 524 | £275.03 (£248.63 to £301.43) | 0.835 (0.828 to 0.842) | £1.42 (−£38.61 to £41.46) | −0.002 (−0.012 to 0.009) | −£35.42 (−£248.71 to £177.87) |
| Usual care | 553 | £273.61 (£248.26 to £298.96) | 0.836 (0.829 to 0.843) | | | |
| Complete-case analysis with MLM: applying unit cost for face-to-face contacts used for telephone contacts | | | | | | |
| Intervention | 413 | £262.52 (£222.73 to £302.32) | 0.836 (0.828 to 0.844) | −£34.81 (−£114.27 to £19.65) | 0.001 (−0.011 to 0.014) | £52.83 (−£222.33 to £327.99) |
| Usual care | 453 | £297.33 (£248.37 to £346.29) | 0.836 (0.828 to 0.843) | | | |
| Complete-case analysis with MLM: medication quantities for SystmOne based on most frequently reported rather than average | | | | | | |
| Intervention | 413 | £259.39 (£219.57 to £299.21) | 0.836 (0.828 to 0.845) | −£29.89 (−£118.22 to £24.25) | 0.001 (−0.011 to 0.014) | £47.92 (−£226.82 to £322.66) |
| Usual care | 453 | £289.28 (£239.48 to £339.08) | 0.836 (0.828 to 0.843) | | | |
| Complete-case analysis with MLM: removal of training costs | | | | | | |
| Intervention | 413 | £245.87 (£208.19 to £283.56) | 0.836 (0.828 to 0.845) | −£37.65 (−£117.50 to £14.97) | 0.001 (−0.011 to 0.014) | £55.67 (−£218.17 to £329.51) |
| Usual care | 453 | £283.52 (£235.40 to £331.64) | 0.836 (0.828 to 0.843) | | | |
| Complete-case analysis with MLM: removal of all intervention costs | | | | | | |
| Intervention | 413 | £213.55 (£176.34 to £250.77) | 0.8436 (0.828 to 0.845) | −£70.28 (−£153.84 to −£18.45) | 0.001 (−0.011 to 0.014) | £88.31 (−£185.50 to £362.12) |
| Usual care | 453 | £283.84 (£235.59 to £332.09) | 0.836 (0.828 to 0.843) | | | |
| MLM: simple imputation of £0 for secondary care per arm and a QALY value 10% higher than the mean QALY value per arm | | | | | | |
| Intervention | 456 | £254.12 (£216.02 to £ 292.21) | 0.842 (0.834 to 0.850) | −£20.19 (−£91.44 to £30.56) | −0.006 (−0.020 to 0.008) | −£96.06 (−£394.16 to £202.04) |
| Usual care | 516 | £274.31 (£231.20 to £317.42) | 0.848 (0.839 to 0.857) | | | |
| MLM: simple imputation of the highest participant cost for secondary care per arm and a QALY value 10% lower than the mean QALY value per arm | | | | | | |
| Intervention | 456 | £330.66 (£273.59 to £387.73) | 0.824 (0.816 to 0.833) | −£322.19 (−£435.12 to −£232.31) | −0.004 (−0.015 to 0.010) | £243.74 (−£39.00 to £526.48) |
| Usual care | 516 | £652.85 (£582.44 to £723.25) | 0.838 (0.821 to 0.836) | | | |

*Adjusted for centre, practice-size and area-level deprivation. In addition, costs were adjusted for general practice IT system and QALYs were adjusted for baseline utility.
†Bootstrapped bias corrected and accelerated confidence intervals.
INMB, incremental net monetary benefit; MLM, multilevel model; QALYs, quality-adjusted life-years; SUR, seemingly unrelated regression.

adjusted costs and QALYs. Compared with usual care, adjusted costs were −£29.99 (95% CI −£109.84 to £22.63) lower and adjusted QALYs were only 0.001 (95% CI −0.011 to 0.014) greater in the intervention arm. The INMB results (£48.01, 95% CI −£225.83 to £321.85) and the CEAC suggest a 63% probability of the intervention being cost-effective when applying the UK's recommended threshold of £20 000 per QALY (figure 1). Together the CEAC and the 95% CIs of the INMB results, which include zero, indicate there is uncertainty in our findings.

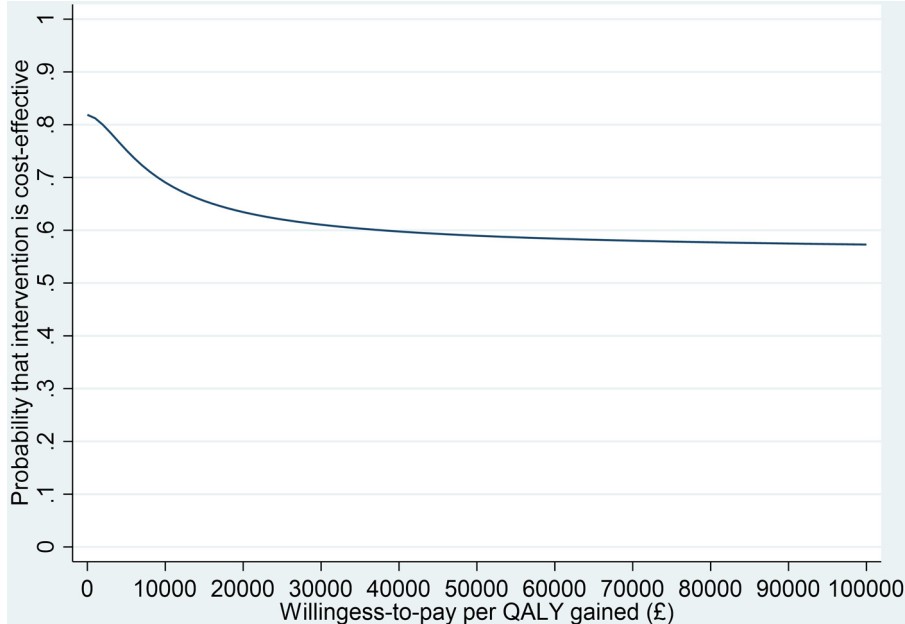

**Figure 1** Cost-effectiveness acceptability curve from an NHS perspective. NHS, National Health Service; QALY, quality-adjusted life year.

## Sensitivity analysis

In general, the sensitivity analyses (table 4) showed similar results to the base-case analysis, with a small positive INMB and wide CIs observed. However, two sensitivity analyses exploring ways to handle missing data approaches (multiple imputation, and assuming zero costs and higher QALYs for missing self-report data) led to small negative INMB and wide CIs. A sensitivity analysis assuming highest costs and lower QALYs for missing self-report data led to a positive INMB as well as wide CIs. Lastly, a £70 (95% CI –£154 to –£18) reduction in cost was observed when intervention costs were excluded.

## Subgroup analysis

More than half of participants in the intervention arm (57.1%, n=236) had completed their 12-month follow-up before 11 March 2020 (table 3, online supplemental material) (beginning of the COVID-19 pandemic). This was the case for a third of participants (37.3%, n=169) in the usual care arm. For the pre-11 March, participant costs in each arm were similar compared with the primary analysis. However, QALYs were slightly lower in the intervention arm, which resulted in a negative INMB (–£156, 95% CI –£730 to £419). Post-11 March 2020, participants in the intervention arm had lower costs and greater QALYs, while participants in the usual care arm had slightly higher costs and fewer QALYs compared with the primary analysis, which led to an overall higher INMB of £640 (95% CI –£1445 to £2727).

## DISCUSSION
### Statement of principal findings

We report on the cost-effectiveness of a manualised and standardised intervention to treat male LUTS in primary care. From a UK NHS perspective, costs and QALYs that had accrued 12 months postconsent were largely similar across both arms. Applying a willingness-to-pay threshold of £20 000 per QALY (as stated in the UK's reference case) resulted in a small positive INMB and a 63% probability of the intervention being cost-effective when compared with usual care. The sensitivity analyses indicated that our base-case results were robust, with the exception of two sensitivity analyses exploring the impact of missing data.

### Strengths and weaknesses

Using EHRs to capture primary care resource use[16] resulted in very high rates (94.9%, n=1022) of complete data. This meant we were able to conduct a complete case base-case analysis with 78.8% (n=413) and 81.9% (n=453) of participants included from the intervention and usual care arm, respectively. Our rates of complete data are higher than those reported in a recent review of missing data in economic evaluation. This found that missing data are very common in trial-based analyses, with studies typically reporting 63% of participants having complete cost-effectiveness data.[17] The authors of the review attributed this issue of missing data to analysts' reliance on self-report questionnaires for measuring resource use and health economic outcomes.

A further advantage of using EHRs was the reduced burden for the participant, especially for those participants in the usual care group who gained little from trial participation. Self-report questionnaires which ask participants to recall details on when they accessed care, what care was provided and by whom, can be cognitively burdensome for participants.[18] Evidence from an earlier large male urology trial which reported low levels of resource use from the patient's perspective,[19] combined

with the concern over participant burden, meant that the current study only took an NHS perspective.

Our EHR data derived from England's two main GP IT systems (EMIS or SystmOne). A noteworthy limitation of using different EHR systems was that they provided different levels of detail. Specifically, the reason for a consultation and the quantities prescribed for a medication could not be extracted from practices which used the SystmOne IT system. In order to reduce any potential bias from using a different IT system, a standardised approach was developed which we have described in our methods. In the regression analysis, we adjusted for IT system and a sensitivity analysis was conducted which did not alter the findings. Future cluster randomised studies using EHR data to capture healthcare resource use should also consider stratifying for the GP IT system within the randomisation process, as well as controlling for this variable.

The COVID-19 pandemic also led to a number of limitations. It was not possible to request secondary care letters for all participants with missing data, because practices were reporting time constraints resulting from the pandemic. Prioritising participants with a urology referral in their EHR data was, therefore, deemed a pragmatic approach. It was assumed the reporting quality was the same for the self-report questionnaires and secondary care letters. One practice site in the intervention arm did not provide EHRs for any of their patients (n=29) taking part in the study because of time and staff constraints resulting from the COVID-19 pandemic.

It was expected that key important and relevant differences between arms for costs and effects would be captured over the 12-month time horizon. Nevertheless, as shown in our post hoc subgroup analysis (table 3, online supplemental material), there was a significant proportion of participants, specifically in the usual care arm, who had their 12-month data collected during the COVID-19 pandemic. It is unclear how the pandemic impacted our analyses. In the wider literature, it has been estimated that there was around a 30% reduction in GP consultations from March 2020 up to June 2020.[20] Therefore, participants whose 12-month follow-up occurred after 11 March 2020 may have reduced healthcare use due to the impact the pandemic had on primary and secondary care services in the UK. Since a larger proportion of usual care participants had their 12-month data collected during the pandemic, it is possible that resource use was underestimated in the usual care arm. Lastly, the subgroup analysis indicated higher costs for the usual care participants whose 12-month follow-up occurred during the pandemic, though the wide CIs suggest this is likely to be due to chance.

Missing data in our study were primarily driven by incomplete self-report questionnaire data for secondary care activity and EQ-5D-5L data, and when these missing data were accounted for using multiple imputation analysis it resulted in the intervention arm having slightly greater costs and lower QALYs than the usual care arm,

indicating the similarity of costs and effects between the two arms.

## Implications for practice

Our study presents the first economic evaluation carried out alongside a large multicentre definitive trial to provide evidence on the value for money of providing a manualised and standardised non-pharmacological intervention in general practice. An earlier smaller trial evaluating self-management for bothersome uncomplicated LUTS did not report on the cost-effectiveness of this type of intervention.[4]

Conservative management for LUTS, as recommended by NICE, is not standardised across UK general practice. Furthermore, an audit by the Royal College of Physicians found most men are not receiving any conservative management for their bothersome LUTS at initial assessments.[21] The consequence of not implementing NICE guidelines could mean men are more likely to endure a reduction in quality of life, continue to call on primary care support due to persistent symptoms, be prescribed prostate medication or be referred to secondary care.

In the UK, an average GP consultation is around 12 minutes,[22] meaning there is limited opportunity for GPs to deliver conservative management, as recommended in NICE guidelines. Our study indicates that it is possible for other primary care healthcare professionals to implement a standardised conservative intervention which provides advice and behavioural techniques to men at a neutral cost.

## CONCLUSION

The cost-effectiveness analysis of the TRIUMPH study showed that there were similar costs and outcomes between the two arms of the trial, indicating that the provision of a manualised and standardised non-pharmacological intervention in general practice can be implemented at a neutral cost. This evidence in conjunction with the improvement shown in the effectiveness study[5], gives support to its implementation within primary care services in the UK.

**Author affiliations**
[1]Population Health Sciences, University of Bristol, Bristol, UK
[2]Department of Surgery & Cancer, Imperial College London, London, UK
[3]Department of Nursing & Midwifery, University of the West of England, Bristol, UK
[4]School of Health Sciences, University of Southampton, Southampton, UK
[5]Faculty of Health and Life Sciences, Oxford Brookes University, Oxford, UK
[6]Bristol Urological Institute, North Bristol NHS Trust Southmead Hospital, Bristol, UK
[7]Faculty of Health Sciences, University of Southampton, Southampton, UK
[8]Brockway Medical Centre, Bristol, UK
[9]NIHR Bristol Biomedical Research Centre, University of Bristol, Bristol, UK
[10]Public and Patient Involvement Representative, Bristol, UK

**Acknowledgements** This study was designed and delivered in collaboration with the Bristol Randomised Trials Collaboration (BRTC), a UKCRC registered clinical trials unit which, as part of the Bristol Trials Centre, is in receipt of National Institute for Health Research CTU support funding. The University of Bristol acted as the Sponsor for this trial and the trial was hosted by the NHS Bristol, North Somerset and South

Gloucestershire Clinical Commissioning Group (CCG). The TRIUMPH Research team acknowledges the support of the National Institute for Health Research Clinical Research Network (NIHR CRN). Study data were collected and managed using REDCap hosted at the University of Bristol.The authors would like to thank all participants, principal investigators and their teams at each of the TRIUMPH study sites for their involvement, and the West of England and Wessex CRNs for their role in the study. The authors would also like to thank the members of the Patient Advisory Group, Trial Steering Committee and Data Monitoring Committee.

**Contributors** MC attests that all listed authors meet authorship criteria and that no others meeting the criteria have been omitted.Conception or design: MC, SN, MJD, AL, NC, MF, LM, HH, SM, MJR, ES, JR, LAR and GT assisted with the study design. JW and JF managed the coordination of the study. MM and JT also assisted with the coordination of the study. Analysis: MC and SN conducted the analysis for the economic evaluation. Drafting the manuscript: MC and SN developed the manuscript for the economic evaluation. All authors contributed to the oversight of the study via the TMG, read and commented on manuscript drafts and approved the final manuscript. MC is the author acting as guarantor for this manuscript.

**Funding** This study was funded by the National Institute for Health Research (NIHR) HTA programme, funding number 16/90/03. The funder had no role in the design and conduct of the study; collection, management, analysis, and interpretation of the data; preparation, review, or approval of the manuscript; and decision to submit the manuscript for publication. Bristol Trials Centre receives National Institute for Health Research CTU Support Funding. This funding has been awarded to support us in developing and supporting NIHR trials. The views expressed are those of Bristol Trials Centre and not necessarily those of the NIHR or the Department of Health and Social Care.

**Competing interests** Prof Marcus Drake reports personal fees from Astellas and Pfizer, outside the submitted work. Dr Jonathan Rees is chair of the Primary Care Urology Society which has received non-promotional sponsorship for annual meetings from Ferring, Astellas, Neotract and IMedicare. He has also received speaker fees from Astellas Pharmaceuticals. Prof Hashim Hashim reports personal fees from Medtronic, Astellas, Allergan and Boston Scientific, outside the submitted work. Prof Athene Lane reports receiving funding for the clinical trials unit (CTU) of which she was co-director, and is currently an active member on the NIHR CTU Standing Advisory Committee. Dr Matthew Ridd has been on several NIHR committees including the Systematic Reviews NIHR Cochrane Incentive Awards, HTA General Committee, Evidence Synthesis Programme Grants Committee, NIHR Incentive Awards Committee and is currently on the Evidence Synthesis Programme Advisory Group. Stephanie MacNeill is an active member of the HTA General Committee.

**Patient and public involvement** Patients and/or the public were involved in the design, or conduct, or reporting, or dissemination plans of this research. Refer to the Methods section for further details.

**Patient consent for publication** Not applicable.

**Ethics approval** This study involves human participants and approval from the NRES North West Preston Ethics Committee (18/NW/0135) was received on 11 April 2018 and applied to all NHS sites who took part in the study. All participants provided their written, informed consent to participation before entering the study. Participants gave informed consent to participate in the study before taking part.

**Provenance and peer review** Not commissioned; externally peer reviewed.

**Data availability statement** Data are available on reasonable request. All data requests should be submitted to the corresponding author for consideration. Access to anonymised data may be granted following review.

**ORCID iDs**
Madeleine Cochrane http://orcid.org/0000-0003-1856-3293
Lucy McGeagh http://orcid.org/0000-0002-9226-5115
Matthew J Ridd http://orcid.org/0000-0002-7954-8823
Luke A Robles http://orcid.org/0000-0003-2882-9868

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
