## [Reviewer comments · BMJ Open]

ARTICLE DETAILS

TITLE (PROVISIONAL)	Cost-effectiveness of a primary healthcare intervention to treat male lower urinary tract symptoms: the TRIUMPH cluster randomised controlled trial
AUTHORS	Cochrane, Madeleine; Drake, Marcus J.; Worthington, Jo; Frost, Jessica; Cotterill, Nikki; Fader, Mandy; McGeagh, Lucy; Hashim, Hashim; Lane, Athene; Macaulay, Margaret; MacNeill, Stephanie; Rees, Jonathan; Ridd, Matthew; Robles, Luke; Sanderson, Emily; Taylor, Gordon; Taylor, Jodi; Noble, Sian

VERSION 1 – REVIEW

REVIEWER	Healey, Andy King's College London, IOPPN
REVIEW RETURNED	04-Sep-2023

GENERAL COMMENTS	Thank you for a clear and well written paper. I have a number of comments/suggested revisions for you to consider. Individually they are fairly minor but there are a few of them. I hope they are helpful. 1. The abstract is generally clear but I was unsure about the conclusion. Does the similarity between costs and QALYs between the arms imply that the intervention can be implemented at low cost? To me the small incremental net benefit of the intervention implies that the net QALY impact of the intervention (accounting for health gains and the health opportunity cost of resource utilisation in both arms) is fairly neutral.2. It would be helpful to have some more clarity in the methods section of the paper on how the EHR and patient self report data were combined to estimate patient level costs.3. Also some more detail would be helpful about how decisions were made regarding which EHR and self-report service contact data were to be attributed to LUTS.4. Please state missing data assumptions regarding complete case analysis and the analysis using multiple imputations (data MCAR, MAR etc.).5. Multi-level modelling is appropriately used to adjust for clustering within the main trial design. In the analysis section the type of model MLM should be stated - e.g. linear mixed effects model using random intercepts (if that was the case) and the number of levels. Also - it would be helpful to report the model outputs for cost and QALYs (perhaps in an appendix) including fixed-effects estimates and the random effects with the intra class correlation (ICC).
--

	6. Please check whether there is an error in table 4 regarding the statistics reported for QALYs. The means and the CIs look suspiciously the same. 7. The results section titled "cost-effectiveness analysis": should the sentence in line 35 page 13 about overlapping confidence intervals actually refer to the INB statistic having a 95% CI that is inclusive of £0? 8. In the same section as the previous point (cost-effectiveness analysis results) - it would be helpful to report that probability that the intervention is cost-effective at the base case threshold here, along with the 95% CIs. 9. If limits on space allow - it would be good to have the CEAC as a figure in the main text or alternatively to report the probability values at different assumed thresholds in table 4. 10. In the sensitivity analysis section of the results section - line 48 page 13. Line reads a "significant reduction". I think this should be statistically significant. But the authors may want to avoid reference to statistical significance and just report confidence intervals (I wouldn't have thought the study was powered to detect differences in costs or QALYs at $\alpha=0.05$?). 11. Lines 56 to 60 on page 18, the authors imply that removal of intervention costs led to observance of an increase in the cost advantage of the intervention that was statistically significant (£70 difference). I'm not convinced by the value of this statement as reducing clinical time involved with delivery could also impact on outcomes. It could also be argued with a degree of plausibility that outside of a well resourced research/trial setting the intervention might be less effective and there might also be additional implementation costs not considered in this study arising from roll-out and scale-up. The authors should be more balanced in what they report here. 12. Statistical software used for the analysis should be reported. And whether the authors followed CHEERS guidelines in their reporting.
--	--

VERSION 1 – AUTHOR RESPONSE

Reviewer: 1		
1. The abstract is generally clear but I was unsure about the conclusion. Does the similarity between costs and QALYs between the arms imply that the intervention can be implemented at low cost? To me the small incremental net benefit of the intervention implies that the net QALY impact of the intervention (accounting for health gains and the health opportunity cost of resource utilisation in both arms) is fairly neutral.	Thank you for suggesting this clarification. We have revised the sentence in our abstract and conclusion.	Abstract: "Costs and QALYs were similar between the two arms at 12 months follow-up. This indicates that the intervention can be implemented in general practice at neutral cost." Page 5 Conclusion: "The cost-effectiveness analysis of the TRIUMPH study showed that there were similar costs and outcomes between the two arms of the trial, indicating that the provision of a manualised and standardised non-pharmacological

		intervention in general practice can be implemented at neutral cost. This evidence in conjunction with the improvement shown in the effectiveness study, gives support to its implementation within primary care services in the UK.” Page 5
2. It would be helpful to have some more clarity in the methods section of the paper on how the EHR and patient self report data were combined to estimate patient level costs.	Thank you, we have added additional information to explain how the EHR and self-report data were combined. Secondary care was predominantly collected via patient self-report. We only used the EHR data to identify secondary referrals. If a participant had a urology referral but had no secondary care self-report data, then we reviewed the participants secondary care letters.	“Secondary care was predominantly collected via patient self-report questionnaires at six and 12-month follow-up, administered either online or via post, were used to capture secondary care LUTS-related healthcare use (outpatient, day case, inpatient and A&E visits). A urology clinician was consulted to identify LUTS-related secondary care visits. EHR data were used to identify secondary referrals. If a participant had a urology referral but had missing or no secondary care self-report data, then we reviewed the participant’s secondary care letters.”
3. Also some more detail would be helpful about how decisions were made regarding which EHR and self-report service contact data were to be attributed to LUTS.	Thank you for this suggestion, we have added detail to explain that a urology clinician was consulted to identify LUTS-related medications and secondary care. All primary care consultations were included in our analysis due to one of the GP systems being unable to extract information regarding the ‘reason for consultation’. We discuss this further in the discussion.	“A urology clinician was consulted to identify LUTS-related prescribed medications and secondary care visits. More specifically, Prescribed medications were extracted from the EHRs if they were coded under the drug criteria: urinary frequency, nocturnal enuresis, incontinence, urinary retention or diuretics. The clinician reviewed the names and doses of the extracted medications and created a list of LUTS-related medications which was applied by the analyst. Similarly, the clinician reviewed a list of the types of secondary care visits reported by the participants and in the primary care letters. The clinician created a list of LUTS-related care and this was applied by the analyst.” Page 9
4. Please state missing data assumptions regarding complete case analysis and the analysis using multiple	We have stated the missing data assumptions in our methods.	“For our primary analysis we assumed our data were missing completely at random. In addition, we tested the assumption that our data were missing at random by

imputations (data MCAR, MAR etc.).		performing multiple imputation by chained equations with predictive mean matching and our SUR model.” Page 12
5. Multi-level modelling is appropriately used to adjust for clustering within the main trial design. In the analysis section the type of model MLM should be stated - e.g. linear mixed effects model using random intercepts (if that was the case) and the number of levels. Also - it would be helpful to report the model outputs for cost and QALYs (perhaps in an appendix) including fixed-effects estimates and the random effects with the intra class correlation (ICC).	Thank you for your suggestion. We have specified our model in further detail. We have also reported our model outputs in the supplementary material, tables 1 and 2. We have updated the table numbers in our text and supplementary material.	“A mixed effects multilevel model (MLM) using random intercepts and assuming a two-level structure” Page 11 “(MLM outputs in Tables 1 and 2, Supplementary Material).” Page 11
6. Please check whether there is an error in table 4 regarding the statistics reported for QALYs. The means and the CIs look suspiciously the same.	Thank you for checking. This is not an error, the mean QALY and CIs for each group were similar as illustrated in the incremental means. There were some slight differences for our analyses and these are recorded. QALYs and CIs were similar for sensitivity analyses which related to adjustments in the costs e.g. removal of intervention/ training costs, using a different unit cost. We have now reported QALYs and CIs to 3 decimal places in Table 4.	Table 4: mean QALYs and CIs are now reported to 3 decimal places. Page 15.
7. The results section titled "cost-effectiveness analysis": should the sentence in line 35 page 13 about overlapping confidence intervals actually refer to the INB statistic having a 95% CI that is inclusive of £0?	Yes, thank you for spotting this. We have reworded the text.	“The positive INMB statistic (£48.01, 95% CI's: -£225.83 to £321.85) indicates the intervention is cost-effective when applying the UK's recommended threshold of £20,000 per QALY, but the 95% confidence intervals include zero indicating there is uncertainty in our findings.” Page 14

8. In the same section as the previous point (cost-effectiveness analysis results) - it would be helpful to report that probability that the intervention is cost-effective at the base case threshold here, along with the 95% CIs.	Thank you for your suggestion, we have reported the probability in the previous sentence.	"The INMB results (£48.01, 95% CI's: -£225.83 to £321.85) and cost-effectiveness acceptability curve (CEAC) suggest a 63% probability of the intervention being cost-effective when applying the UK's recommended threshold of £20,000 per QALY (Figure 1). Together the CEAC and the 95% confidence intervals of the INMB results, which include zero, indicates there is uncertainty in our findings." Page 14
9. If limits on space allow - it would be good to have the CEAC as a figure in the main text or alternatively to report the probability values at different assumed thresholds in table 4.	Our CEAC is presented in Figure 1, which was uploaded in our original submission to the BMJ Open. The intention is that Figure 1 will be placed within the main manuscript. We have also provided the probability value (63%) for our base case analysis in the text of the main manuscript.	No change needed
10. In the sensitivity analysis section of the results section - line 48 page 13. Line reads a "significant reduction". I think this should be statistically significant". But the authors may want to avoid reference to statistical significance and just report confidence intervals (I wouldn't have thought the study was powered to detect differences in costs or QALYs at alpha=0.05?).	Thank you, we have adjusted the wording of this sentence.	"Lastly, a £70 (95% CIs: -£154 to -£18) reduction in cost was observed when intervention costs were excluded." Page 14
11. Lines 56 to 60 on page 18, the authors imply that removal of intervention costs led to observance of an increase in the cost advantage of the intervention that was statistically significant (£70 difference). I'm not convinced by the value of this statement as reducing clinical time involved with delivery could	Thank you for this comment, on reflection to ensure a more balanced approach, we have deleted these lines.	DELETED: "Not including the intervention costs (£39) in the analysis led to a significant difference in costs of £70 in favour of the intervention arm indicating that if costs of implementing the intervention could be reduced further for example by reducing the number of follow up contacts, as proposed in the effectiveness evaluation, then the rollout of this

also impact on outcomes. It could also be argued with a degree of plausibility that outside of a well resourced research/trial setting the intervention might be less effective and there might also be additional implementation costs not considered in this study arising from roll-out and scale-up. The authors should be more balanced in what they report here.		intervention could potentially lead to greater cost savings to the NHS" Page 19
12. Statistical software used for the analysis should be reported. And whether the authors followed to CHEERS guidelines in their reporting.	Thank you we have added this information.	"The analysis is reported according to the Consolidated Health Economic Evaluation Reporting Standards (CHEERS) guidelines(5)." Page 7 "STATA version 16.1 was used for all analyses" Page 10

VERSION 2 – REVIEW

REVIEWER	Healey, Andy King's College London, IOPPN
REVIEW RETURNED	04-Dec-2023
GENERAL COMMENTS	Thank you for addressing the minor comments I raised on the earlier version of the manuscript.